# A simple low-latency real-time certifiable quantum random number generator

Yanbao Zhang [1,2,4 ✉], Hsin-Pin Lo [1,4 ✉], Alan Mink[3], Takuya Ikuta [1], Toshimori Honjo[1], Hiroki Takesue[1] & William J. Munro [1,2]

Quantum random numbers distinguish themselves from others by their intrinsic unpredictability arising from the principles of quantum mechanics. As such they are extremely useful in many scientific and real-world applications with considerable efforts going into their realizations. Most demonstrations focus on high asymptotic generation rates. For this goal, a large number of repeated trials are required to accumulate a significant store of certifiable randomness, resulting in a high latency between the initial request and the delivery of the requested random bits. Here we demonstrate low-latency real-time certifiable randomness generation from measurements on photonic time-bin states. For this, we develop methods to certify randomness taking into account adversarial imperfections in both the state preparation and the measurement apparatus. Every 0.12 s we generate a block of 8192 random bits which are certifiable against all quantum adversaries with an error bounded by $2^{-64}$. Our quantum random number generator is thus well suited for realizing a continuously-operating, high-security and high-speed quantum randomness beacon.

[1] NTT Basic Research Laboratories, NTT Corporation, Atsugi, Kanagawa, Japan. [2] NTT Research Center for Theoretical Quantum Physics, NTT Corporation, Atsugi, Kanagawa, Japan. [3] National Institute of Standards and Technology, Gaithersburg, MD, USA. [4]These authors contributed equally: Yanbao Zhang, Hsin-Pin Lo. ✉email: yanbao.zhang.xf@hco.ntt.co.jp; hsinpin.lo.cn@hco.ntt.co.jp

Quantum mechanics is well known to offer many opportunities for generating genuine randomness that is unpredictable by any reference[1–3]. This unpredictability can be proven based only on measurement observations and a few assumptions. Therefore, the randomness generated according to quantum mechanics is certifiable. The simplest example involves measuring a two-level quantum system (a qubit) prepared in an equal superposition of its two levels. However, its proper working and certifiability rely on the trust of both the quantum state prepared and the measurement performed. This scheme is thus device-dependent[2,3]. On the other hand, there are also device-independent schemes that do not require any trust on the inner working of the employed quantum devices[4,5]. Unfortunately, it is difficult to realize such a scheme for practical use with excellent performance as it requires a loophole-free Bell test[6–11]. Consequently, the randomness-generation rates achieved are extremely low with a high latency from the beginning of the experiment to the output of the certified random bits[12–15]. The natural question then is whether we can reduce the trust required by the above simple scheme while avoiding the difficulties inherent in the device-independent approach.

In this work we explore a simple practical scheme for the realization of a low-latency real-time certifiable quantum random number generator (QRNG). The simple scheme works ideally as follows: At each trial a horizontally polarized single photon is emitted from a source, and then measured randomly along either the $X$-basis (diagonal/anti-diagonal polarization basis) to generate a random bit or the $Z$-basis (horizontal/vertical polarization basis) to verify the prepared state. This scheme is motivated by that for entanglement-based quantum key distribution (QKD)[16,17], where one basis is used to generate secret keys and other bases are used to estimate the prepared state. Random bits or secret keys can be certified since measurement outcomes allow us to bound the correlation between the prepared state and the side information of an adversary known as Eve[18].

The above ideal scheme has been well studied in the literature[19,20]. However, in order to make the resulting QRNG practical, we need to consider the imperfections in its implementations and show the robustness of randomness generation against those imperfections. First, single-photon sources are not easily accessible and as for QKD[18], weak optical pulses are usually employed. Even if a single-photon source is available, it is still generally difficult to produce a particular quantum state with high accuracy. Second, it is difficult in an experiment to perform measurements precisely along both the $X$-basis and $Z$-basis, as one basis tends to be more precise than the other. Third, the basis choice at a trial is usually made by a pseudo or physical random number generator. This means that the probabilities of selecting the $X$-basis and $Z$-basis, denoted as $P_X$ and $P_Z$, can only be bounded but not exactly known. Furthermore, in the adversarial scenario Eve could manipulate these imperfections. These adversarial imperfections must be addressed together to reliably certify randomness which currently has not been done.

Here we develop a method to guarantee the proper working and security of our QRNG in the presence of those above adversarial imperfections. For this, we require a lower bound $q_{1,\mathrm{lb}}$ on the single-photon probability in a practical photon source (such as a weak laser pulse in the absence of a phase reference), an upper bound $\delta$ on the misalignment angle between the $X$-basis and $Z$-basis, and both a lower and an upper bounds on the imbalance between the probabilities $P_X$ and $P_Z$ given by $\tau = (P_X - P_Z)/2$. We emphasize that except the above bounds which characterize the adversarial imperfections, our method does not need any other information about the state prepared or measurements performed. In this sense, our QRNG works in a semi-device-independent way. The values of the above imperfection bounds can be obtained by

calibrating the photon source and measurement apparatuses in real time. We allow Eve to manipulate the state prepared or measurements performed as long as these manipulations satisfy the above imperfection bounds. Our method is of excellent finite-data efficiency, thus enabling low-latency real-time randomness generation. Specifically, we experimentally demonstrate that every 0.1 s a sufficient amount of entropy with respect to the quantum (or classical) side information of Eve is certified such that a block of 8192 (or $2 \times 8192$) random bits is generated with a certified error bounded by $2^{-64}$ and with an extraction time of 0.02 s (or 0.04 s).

## Results

**Outline**. In what follows, we first introduce the setup of the problem and the main idea of our method for certifying randomness with the adversarial imperfections discussed above. Our method works in the presence of both the classical and quantum side information of Eve. We then illustrate the performance of our method with simulations, showing the advantage of Eve with an access to quantum side information. Finally, we present our experimental realization of a simple low-latency real-time QRNG enabled by our method.

**Setup of the problem**. To generate random bits, we consider an experiment with a sequence of $n$ repeated trials. These trials are not necessarily independent or identical. We denote the input (basis choice) and the output (measurement outcome) at the $k$'th trial by the random variables $I_k$ and $O_k$, respectively. The inputs and outputs of the experiment are then $\mathbf{I}_n = (I_k)_{k=1}^n$ and $\mathbf{O}_n = (O_k)_{k=1}^n$. The amount of randomness in the outputs relative to both the inputs and Eve is quantified by the smooth conditional min-entropy $H_{\min}^{\epsilon_s}(\mathbf{O}_n | \mathbf{I}_n, \text{Eve})$, where $\epsilon_s$ is the smoothness error[21]. We consider two alternative smooth conditional min-entropies $H_{\min,\mathrm{c}}^{\epsilon_s}(\mathbf{O}_n | \mathbf{I}_n, \text{Eve})$ and $H_{\min,\mathrm{q}}^{\epsilon_s}(\mathbf{O}_n | \mathbf{I}_n, \text{Eve})$ in the presence of the classical and quantum side information of Eve, respectively. The ability of Eve to access quantum side information (which is stored in a quantum system $\mathsf{E}$) as compared with classical side information (which is stored in a classical, random variable $E$) allows attacks that can take advantage of long-term quantum memories[22,23] correlated in a quantum manner with the quantum devices used for the state preparation in the experiment. Our goal is to bound the smooth conditional min-entropies $H_{\min,\mathrm{c}}^{\epsilon_s}(\mathbf{O}_n | \mathbf{I}_n, \text{Eve})$ and $H_{\min,\mathrm{q}}^{\epsilon_s}(\mathbf{O}_n | \mathbf{I}_n, \text{Eve})$ from below.

For certifying the randomness in the outputs $\mathbf{O}_n$ relative to the inputs $\mathbf{I}_n$ and Eve, we must assume that the outputs $\mathbf{O}_n$ are kept private and not accessible to Eve. We allow Eve to hold classical or quantum side information about the state prepared at a trial. At the same time, we allow Eve to manipulate the distribution of the possible inputs and the specific forms of the associated measurements at the trial, as long as these manipulations satisfy the prespecified imperfection bounds. We assume that by manipulations Eve can access classical side information but not quantum side information about the measurement performed. The method to be presented allows classical correlations between Eve's side information about the state prepared and Eve's partial knowledge of the input and measurement used at each trial. That is, the state prepared can be classically correlated with the input selected or the measurement performed. We emphasize that our method cannot be applied in the case where at each trial Eve's side information about the state is correlated in a quantum manner with Eve's partial knowledge of the input and measurement. Moreover, although we allow Eve to manipulate the input distribution, we assume that before a trial Eve has no perfect

knowledge of which specific input to be selected at the trial. This assumption is required for security analysis; otherwise, Eve can deterministically forecast the output of the trial, and it would be therefore impossible to certify randomness[24].

**Main idea of our method**. For certifying randomness with respect to classical and quantum side information, we construct probability estimation factors (PEFs)[25,26] and quantum estimation factors (QEFs)[27,28], respectively. Both a PEF and a QEF are non-negative functions of the input $I$ and output $O$ of a trial, denoted by $F_c(I, O)$ and $F_q(I, O)$. The key observation is that the smooth conditional min-entropies $H_{\min, c}^{\epsilon_s}(\mathbf{O}_n | \mathbf{I}_n, \text{Eve})$ and $H_{\min, q}^{\epsilon_s}(\mathbf{O}_n | \mathbf{I}_n, \text{Eve})$ can be bounded from below, once we know the respective products $\prod_{k=1}^{n} F_c(i_k, o_k)$ and $\prod_{k=1}^{n} F_q(i_k, o_k)$. Here, $i_k$ and $o_k$ are the observed values of the input and output at the $k$'th trial. This key observation can be formalized by Theorem 1 and Theorem 2 in the "Methods" section. We emphasize that PEFs and QEFs can use the result of each trial for both verifying and accumulating randomness. Both PEFs and QEFs have been constructed for certifying device-independent randomness[15,25–28]. In this work, we develop methods to construct PEFs and QEFs for the scenario of our interest. In particular, the PEFs and QEFs constructed are adapted to the adversarial imperfections in both the state source and the measurement apparatus. Both PEFs and QEFs have the advantage that significantly less data is required in order to certify a fixed amount of randomness. Details for constructing PEFs and QEFs are discussed in the "Methods" section.

After certifying the amount of randomness, we run the randomness extractor developed in ref. [29] with extractor error $\epsilon_x = \epsilon - \epsilon_s$ in order to generate random bits which are within distance of $\epsilon > \epsilon_s$ from uniform. The distance $\epsilon$ is termed the soundness error. For the results presented in this work, we set the smoothness error and the extractor error to be $\epsilon_s = 0.8\epsilon$ and $\epsilon_x = 0.2\epsilon$.

**Advantage of quantum adversaries over classical adversaries**. We illustrate with simulations the performance of our method in the asymptotic limit, so that one can see the expected behavior of our QRNG scheme. When the trials are identical and $n$ approaches infinity, the amount of randomness certified by our method increases linearly with $n$. The increasing rate (per trial) is called the asymptotic randomness-generation rate. The rates in the presence of classical and quantum side information, $R_c$ and $R_q$, certified by our method are optimal (see refs. [25,27] for general proofs). We can quantify $R_c$ and $R_q$ as functions of the depolarization noise $d$ (as defined in the caption of Fig. 1). The results presented in Fig. 1 clearly indicate that Eve's access to quantum side information as compared with classical side information results in a reduction of the randomness-generation rate. Such a reduction is an important yet unquantified advantage to Eve.

**Experimental realization of a simple low-latency real-time QRNG**. To realize a QRNG, we perform measurements on photonic time-bin states, where the quantum information is encoded into the superposition of two different temporal positions (time bins) of an optical pulse. The two time bins are usually called the early and late time bins denoted by $t_e$ and $t_l$. Time-bin encoding has been widely used especially in fiber-based quantum communication systems[30]. The advantage of time-bin encoding lies in that both the state source and the measurement apparatus required are easily packaged onto a chip, which is an important factor to consider for practical QRNG use.

To produce randomness, at each trial we attempt to prepare the time-bin qubit state $|1_{t_e}\rangle \otimes |0_{t_l}\rangle$, where $|j_t\rangle$ represents the

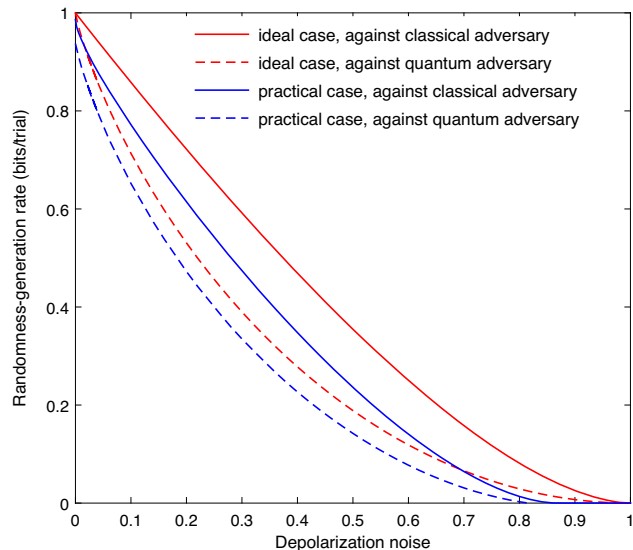

**Fig. 1 Asymptotic randomness-generation rates $R_c$ and $R_q$ as functions of the depolarization noise $d$.** For illustration purpose, here we simulate the result at a trial according to either the $X$-basis or $Z$-basis measurement on the depolarized single-photon state $(1 - d/2)|0\rangle\langle 0| + d/2|1\rangle\langle 1|$, where $|0\rangle$ and $|1\rangle$ are the two eigenstates (in the single-photon subspace) of the $Z$-basis measurement and $d$ quantifies the depolarization noise. At each trial the $X$-basis measurement is selected with probability $P_X = 0.9999$, and so the imbalance $\tau = (P_X - P_Z)/2$ is exactly known. Our method can certify randomness without assuming that the state and measurements are fully characterized. Instead, our method requires only an upper bound $\delta$ on the misalignment angle between the two measurement bases and a lower bound $q_{1,\text{lb}}$ on the probability of a single photon in a practical photon source. For the ideal case (the red curves), we set $q_{1,\text{lb}} = 1$ and $\delta = 0$, while for the practical case (the blue curves), we set $q_{1,\text{lb}} = 0.95$ and $\delta = 5°$.

$j$-photon state located at the time bin $t \in \{t_e, t_l\}$. After passing it through an unbalanced Mach–Zehnder interferometer (MZI), we measure the time-bin qubit, as depicted in Fig. 2. The difference in photon transit time between the two unbalanced paths of the MZI matches the separation between $t_e$ and $t_l$. Therefore, a photon can come out from the MZI at the early, middle and late time bins denoted by $t_e'$, $t_m'$ and $t_l'$, respectively. If the photon comes out at $t_e'$ or $t_l'$, then the $Z$-basis (time-bin basis) is passively selected. In this case, the arrival time indicates the measurement outcome. If the photon comes out at $t_m'$, then the $X$-basis (superposition basis) is passively selected. In this case, the two output ports of the MZI indicate which measurement outcomes are observed. Note that if the first beam splitter in the MZI has the 50:50 splitting ratio, the two measurement bases are uniformly randomly selected. In this sense, the first beam splitter in the MZI acts effectively as a physical but uncertified random number generator[31].

In practice, the source emits zero photon with a non-zero probability at each trial, and threshold detectors (which cannot resolve photon number) of finite efficiency are employed. Moreover, a photon can be lost over the transmission from the source to the detectors. Therefore, not all trials have detector clicks. For security analysis, we assume that the trials with detector clicks are a fair sample of all trials. Accordingly, no-click events do not affect the security analysis of randomness generation but only the rate and latency achieved in practice.

Now for certifying randomness, we must take into account the adversarial imperfections in our setup. Neither of the two beam splitters, BS1 and BS2, in the MZI has the ideal 50:50 splitting

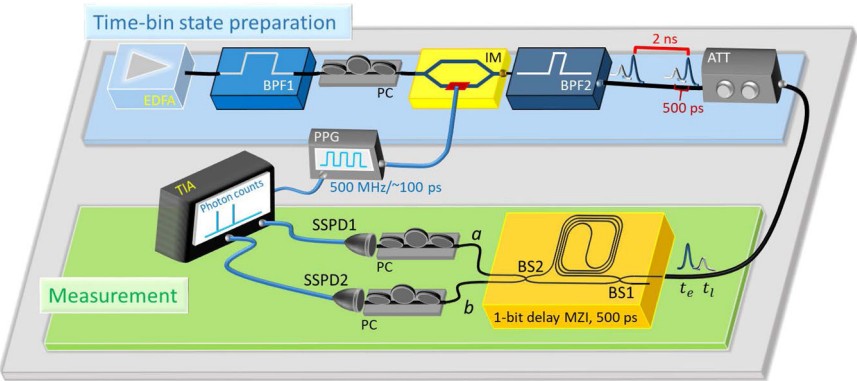

**Fig. 2 Schematic diagram of our experimental setup.** First, the time-bin state $|1_{t_e}\rangle \otimes |0_{t_l}\rangle$, where there is a single photon at the early time bin $t_e$ and no photon at the late time bin $t_l$, is prepared in the ideal case. Second, after passing over an unbalanced Mach–Zehnder interferometer (MZI), the optical pulse is detected by two superconducting nanowire single-photon detectors (SSPDs). The MZI is composed of two beam splitters, BS1 and BS2, and has two output ports $a$ and $b$. In the diagram the abbreviations have the following meanings. EDFA: erbium-doped fiber amplifier, BPF: band-pass filter, PC: polarization controller, IM: intensity modulator, ATT: optical attenuator, PPG: pulse pattern generator, TIA: time-interval analyzer. See the "Methods" section for details.

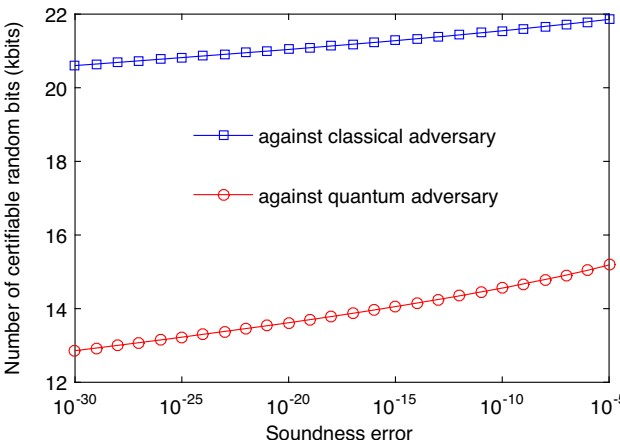

**Fig. 3 Trade-off between the soundness error $\epsilon$ and the expected number of random bits certifiable from the measurement outcomes observed every 0.1 s runtime.** The results in the presence of classical and quantum side information are shown as the blue and red curves, respectively.

ratio. In addition, the two detectors at the output ports $a$ and $b$ may have different efficiencies $\eta_a$ and $\eta_b$. These facts induce not only an imbalance between the probabilities $P_X$ and $P_Z$ of selecting the $X$-basis and $Z$-basis but also a misalignment between the two bases. Based on a calibration of our measurement apparatus, we found that the splitting ratios of BS1 and BS2 are 53.8:46.2 and 46.9:53.1, respectively, and that the ratio $\eta_a : \eta_b$ is 1.024:1. Consequently, the imbalance $\tau = (P_X - P_Z)/2$ and misalignment $\delta$ satisfy the conditions $|\tau| \leq 0.041$ and $\delta \leq 3.565°$. Moreover, we estimated that the single-photon component of the optical pulse contributes at least 99.3% of all click events. More details behind the above characterizations are available in Supplementary Note 3. Accordingly, we conservatively assume that $|\tau| \leq 0.06$, $\delta \leq 6°$, and $q_{1,\text{lb}} = 0.98$ in our security analysis, specifically, for constructing PEFs and QEFs to guarantee certifiable randomness generation.

Based on a set of calibration data, we estimated the expected number, $k_{\exp}$, of random bits certifiable every 0.1 s runtime at a soundness error $\epsilon$ varying from $10^{-5}$ to $10^{-30}$. The dependence of $k_{\exp}$ on $\epsilon$ in the presence of either quantum or classical side information is illustrated in Fig. 3. As expected fewer number of random bits can be certified with respect to quantum side

information than with respect to classical side information. However, the number of certifiable bits in each situation is not significantly affected by the soundness error in the range considered.

We finally consider a request for a block of 8192 (or $2 \times 8192$) random bits in the presence of quantum (or classical) side information and with soundness error bounded by $2^{-64} \approx 5.42 \times 10^{-20}$. The results in Fig. 3 strongly suggest that our QRNG can successfully fulfill the request every 0.1 s runtime. Indeed, the success probability is estimated to be at least $1 - 2^{-380}$ (or $1 - 2^{-478}$) in the presence of quantum (or classical) side information (see Supplementary Note 4 for details). We further demonstrate this repeated fulfillment in experiment. For this, before the experiment we fixed the PEF and QEF used, as well as several other parameters used in our security analysis, based on the above calibration data (see Supplementary Note 4). Then we ran the experiment for 420 s and processed the data block obtained every 0.1 s runtime successively. For each data block, we certified a lower bound on the number of random bits extractable with soundness error $2^{-64}$ and with respect to either quantum or classical side information. If the certified lower bound exceeds the request threshold, the instance of our QRNG succeeds. Conditional on success, we run the randomness extractor developed in[29] to generate the final random bits. The randomness extractor is seed-efficient and requires an additional processing time: for extracting 8192 (or $2 \times 8192$) random bits it takes 0.02 s (or 0.04 s), respectively. Totally we ran 4200 instances of our QRNG. The analysis results summarized in Fig. 4 show the success of each instance.

## Discussion

In conclusion, we demonstrate a simple low-latency real-time certifiable quantum random number generator (QRNG). The generator is based on the measurement of a weak optical pulse with an unbalanced Mach-Zehnder interferometer. By developing an efficient security-analysis method, genuine randomness can be certified and then generated with a low latency from every short block of experimental data even at an extremely high security level and even considering adversarial imperfections in our experimental setup. Further, the implementation of randomness extraction allows real-time performance to be achieved. Our QRNG is thus well suited for realizing a continuously-operating, high-security, and high-speed quantum randomness beacon.

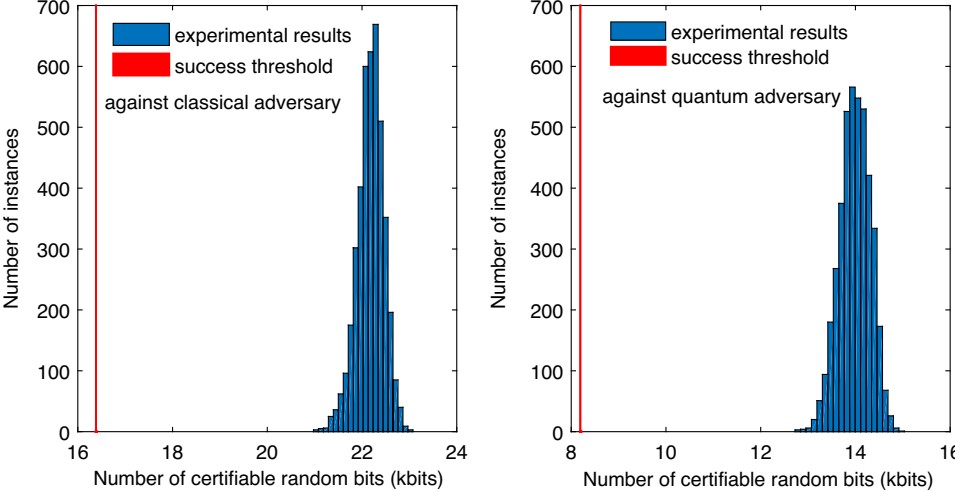

**Fig. 4 Histogram of the numbers of random bits certifiable with soundness error $2^{-64}$ from 4200 instances of our QRNG: the left and right panels are for certifying randomness with respect to classical and quantum side information, respectively.** Each instance of our QRNG uses a data block obtained in 0.1 s runtime. In each panel, the experimental results are shown as blue bars, while the success threshold is shown as the red line.

Our security analysis considers both quantum and classical side information. Our security certificate is resistant to the adversarial imperfections in both the state source and the measurement apparatus, in contrast to those certificates achieved in previous works[20,32–34] where either the adversarial imperfections in the source or those in the measurement apparatus are considered. Moreover, our method exhibits unsurpassed finite-data efficiency. As certifying smooth conditional min-entropies is also the central task for quantum key distribution (QKD), we envision that our method can be extended to improve the finite-data efficiency of QKD. In the future work, we will address the details required for this extension.

## Methods

**Outline**. Here we provide details of our experimental setup for realizing a simple low-latency real-time certifiable quantum random number generator. We also introduce the general framework of probability estimation (or quantum probability estimation) for certifiable randomness generation in the presence of classical (or quantum) side information. Further, we discuss the details of implementing these general frameworks in the presence of the adversarial imperfections considered in both the state source and the measurement apparatus.

**Experimental implementation**. Our experimental setup is shown in Fig. 2. To generate time-bin states, amplified spontaneous emission from an erbium-doped fiber amplifier (EDFA), which has a broad spectrum and thus can be regarded as inherently dephased, is used as a light source. After reducing its bandwidth by a band-pass filter (BPF1) of $1551.1 \pm 1.2$ nm, the light from the EDFA is sent into an intensity modulator (IM) to generate (in the ideal case) the time-bin qubit state consisting of the single-photon pulse $|1_{t_e}\rangle$ and the vacuum pulse $|0_{t_l}\rangle$. A pulse pattern generator (PPG) is used to modulate the IM at a repetition rate of 500 MHz using a pulse of width approximately 100 ps. The same modulation signal is also sent to the time-interval analyzer (TIA), to synchronize the IM and TIA. A BPF2 of $1551.1 \pm 0.44$ nm is then used to further surpress the noise outside of the bandwidth. With the help of an optical attenuator (ATT), we then adjust the average photon number per pulse to a value of approximately 0.0035. Finally, we launch the time-bin pulse into an unbalanced Mach–Zehnder interferometer (MZI), which is fabricated using planar lightwave circuit technologies[35]. The path difference of the unbalanced MZI is 500 ps, the same as the time separation between the early and late time bins. The insertion loss of the MZI is approximately 2.0 dB. The photons from the output ports of the MZI are detected by two superconducting nanowire single-photon detectors (SSPDs), where the detection events are recorded by the TIA. The system detection efficiency of each SSPD is about 59%, and the dark count rate of each SSPD is less than $40~\text{s}^{-1}$. A few polarization controllers (PCs) are inserted before the IM and SSPDs in order to adjust the polarization of photons. We measure that roughly 470,000 trials with detector clicks are generated per second.

**Certifiable randomness generation in the presence of classical side information**. To certify randomness with respect to the classical side information of Eve, we apply the framework of probability estimation as developed in refs. [25,26]. For this, we need to characterize each trial of the experiment by a classical model. In the scenario of our interest, the model is adapted to the adversarial imperfections considered. Given the model, we construct probability estimation factors (PEFs) which can certify randomness with respect to classical side information. Below we first introduce the concepts of classical models and PEFs, and then present the main result of probability estimation for randomness generation.

Let us focus on a generic trial in the experiment with an input $I$ and an output $O$. We omit the trial index for generic trials. As is conventional, we denote a random variable and its possible value by an upper-case letter in regular math font and the corresponding lower-case letter. The classical side information $E$ of Eve can be correlated with the trial input $I$ and trial output $O$. This correlation is described by a joint probability distribution $\mathbb{P}(I, O, E)$. However, in practice we cannot access the classical side information $E$ held by Eve. Therefore, we can characterize only the distribution of $I$ and $O$ conditional on each possible value $e$ of $E$, denoted by $\mathbb{P}(I, O|E = e)$. The set of conditional distributions $\mathbb{P}(I, O|E = e)$, for all possible $e$, achievable at a trial is defined to be the classical model $\mathcal{C}$ for the trial. For simplicity we make the condition on Eve's classical side information implicit in the rest of the paper, and so the classical model $\mathcal{C}$ specifies the set of probability distributions $\mathbb{P}(I, O)$ achievable at a trial. To certify randomness in the output $O$ conditional on the input $I$ and on the classical side information $E$, we consider a class of non-negative functions $F_c : (i, o) \mapsto F_c(i, o)$, called PEFs for the classical trial model $\mathcal{C}$. A PEF with a positive power $\beta_c$ is a non-negative function $F_c : (i, o) \mapsto F_c(i, o)$ which satisfies the PEF inequality

$$\sum_{i,o} \mathbb{P}(I = i, O = o) F_c(i, o) \mathbb{P}(O = o|I = i)^{\beta_c} \leq 1 \qquad (1)$$

at each probability distribution $\mathbb{P}(I, O)$ in the classical trial model $\mathcal{C}$. We have two remarks on the constructions of the classical trial model and the corresponding PEFs as follows: First, when Eve's classical side information about the state is classically correlated with Eve's partial knowledge of the input and measurement at a trial, the classical trial model will become the convex closure of the model $\mathcal{C}$ as introduced above. Second, according to Lemma 14 of ref. [26], a PEF with power $\beta_c$ for the model $\mathcal{C}$ is also a PEF with the same power for the convex closure of $\mathcal{C}$. In view of the above two remarks, probability estimation automatically handles the classical correlation between Eve's classical side information about the state and Eve's partial knowledge of the input and measurement at a trial.

The number of near-uniform random bits extractable from the outputs $\mathbf{O}_n$ given the inputs $\mathbf{I}_n$ and the classical side information $E$ of Eve is quantified by the classical smooth conditional min-entropy $H_{\min, c}^{\epsilon_s}(\mathbf{O}_n|\mathbf{I}_n, \text{Eve})$[21]. Here, the smoothness error $\epsilon_s$ measures the total-variation distance between the actual distribution and an ideal distribution of $\mathbf{I}_n$, $\mathbf{O}_n$ and $E$ (see Definition 9 of ref. [26]). Suppose that each trial of an experiment is characterized by the classical model $\mathcal{C}$. Denote the PEF with power $\beta_c$ at the $k$'th trial by $F_{c,k}$, which is a function of $I_k$ and $O_k$, and let the variable $T_{c,n}$ be the product of PEFs up to the $n$'th trial, that is, $T_{c,n} = \prod_{k=1}^{n} F_{c,k}$. In practice, the input at a trial is independent of the outputs of the previous trials conditionally on the classical side information $E$ and the inputs of the previous trials. Under this conditional-independence condition, probability estimation can certify randomness with respect to classical side information according to the following theorem:

*Theorem 1* (Theorem 1 of ref. [26]): Let $1 \geq \kappa$, $\epsilon_s > 0$ and $1 \geq p \geq 1/|\mathrm{Rng}(\mathbf{O}_n)|$, where $|\mathrm{Rng}(\mathbf{O}_n)|$ is the number of possible outputs after $n$ trials. Define $\Phi$ to be the event that $T_{c,n} \geq 1/(p^{\beta_c} \epsilon_s)$. For each joint probability distribution $\mathbb{P}(\mathbf{I}_n, \mathbf{O}_n, E)$, either the probability of the event $\Phi$ is less than $\kappa$ or the classical smooth conditional min-entropy, when the event $\Phi$ happens, satisfies

$$H_{\min,c}^{\epsilon_s}(\mathbf{O}_n|\mathbf{I}_n, \text{Eve}, \Phi) \geq -\log_2(p) + \frac{1+\beta_c}{\beta_c}\log_2(\kappa). \tag{2}$$

The event $\Phi$ can be interpreted as the event that the experiment succeeds. When the experiment succeeds, we compose the classical smooth conditional min-entropy bound in Eq. (2) with a classical-proof strong extractor of error $\epsilon_x$ (in total-variation distance), in order to obtain random bits which are within soundness error (in total-variation distance) $\epsilon = \epsilon_s + \epsilon_x$ from uniform in the presence of classical side information. See Sect. IV C of ref. [25] for the details of the end-to-end randomness generation. Note that an extractor is strong if the joint of its output and the seed is nearly uniform, while an extractor is classical-proof if it works in the presence of the classical side information. In our experiment, we used Trevisan's extractor[36] as implemented by Mauerer, Portmann, and Scholz[29], which we refer to as the TMPS extractor. The TMPS extractor is an efficient classical-proof strong extractor that requires few seed bits[29,36]. The way of running the TMPS extractor for our case is the same as for the case of device-independent randomness generation with respect to classical side information studied in refs. [13,25].

### Certifiable randomness generation in the presence of quantum side information.

To certify randomness with respect to the quantum side information of Eve, we apply the framework of quantum probability estimation as developed in refs. [27,28]. For this, we need to characterize each trial of the experiment by a quantum model. In the scenario of our interest, the model is adapted to the adversarial imperfections considered. Given the model, we construct quantum estimation factors (QEFs) which can certify randomness with respect to quantum side information. Below we first introduce the concepts of quantum models and QEFs, and then present the main result of quantum probability estimation for randomness generation.

Consider a generic experimental trial which has a classical input $I$ and a classical output $O$. Suppose that Eve holds a quantum system $\mathsf{E}$, which carries the quantum side information about the experiment. So, the quantum system $\mathsf{E}$ is correlated with the trial input $I$ and trial output $O$. The correlation between $\mathsf{E}$ and $(I, O)$ can be described by a classical-quantum state

$$\rho_{IO\mathsf{E}} = \sum_{i,o}|i,o\rangle\langle i,o| \otimes \rho_{\mathsf{E}}(i,o), \tag{3}$$

where $\rho_{\mathsf{E}}(i,o)$ is the sub-normalized state of $\mathsf{E}$ conditional on $I = i$ and $O = o$. The trace $\mathrm{Tr}(\rho_{\mathsf{E}}(i,o))$ is the probability of observing that $I = i$ and $O = o$ at a trial. Since the system $\mathsf{E}$ is inaccessible by us, we consider the set of all the possible classical-quantum states that can occur at the end of the trial. This set is defined to be the quantum model $\mathcal{Q}$ for the trial. We characterize the unpredictability of an output $c$ given both an input $i$ and the quantum side information in $\mathsf{E}$ by the sandwiched Rényi power $R_{\alpha_q}(\rho_{\mathsf{E}}(i,o)|\rho_{\mathsf{E}}(i))$ expressed as

$$\mathrm{Tr}\left(\left(\rho_{\mathsf{E}}(i)^{-\beta_q/2\alpha_q}\rho_{\mathsf{E}}(i,o)\rho_{\mathsf{E}}(i)^{-\beta_q/2\alpha_q}\right)^{\alpha_q}\right), \tag{4}$$

where $\beta_q > 0$ is a free parameter, $\alpha_q = 1 + \beta_q$, and $\rho_{\mathsf{E}}(i) = \Sigma_o\rho_{\mathsf{E}}(i,o)$. To certify randomness in the output $O$ conditional on the input $I$ and on the quantum side information in $\mathsf{E}$, we consider a class of non-negative functions $F_q: (i,o) \mapsto F_q(i,o)$, called QEFs for the quantum trial model $\mathcal{Q}$. A QEF with a positive power $\beta_q$ is a non-negative function $F_q: (i,o) \mapsto F_q(i,o)$ which satisfies the QEF inequality

$$\sum_{i,o}F_q(i,o)R_{\alpha_q}(\rho_{\mathsf{E}}(i,o)|\rho_{\mathsf{E}}(i)) \leq 1 \tag{5}$$

at all states $\rho_{IO\mathsf{E}}$ in the quantum trial model $\mathcal{Q}$. We have two remarks on the constructions of the quantum trial model and the corresponding QEFs as follows: First, when Eve's quantum side information about the state is classically correlated with Eve's partial knowledge of the input and measurement at a trial, the quantum trial model will become the convex closure of the model $\mathcal{Q}$ as introduced above. Second, according to Property 2 of ref. [28], a QEF with power $\beta_q$ for the model $\mathcal{Q}$ is also a QEF with the same power for the convex closure of $\mathcal{Q}$. In view of the above two remarks, quantum probability estimation automatically handles the classical correlation between Eve's quantum side information about the state and Eve's partial knowledge of the input and measurement at a trial.

The number of near-uniform random bits extractable from the outputs $\mathbf{O}_n$ given the inputs $\mathbf{I}_n$ and the quantum side information carried by the system $\mathsf{E}$ of Eve is quantified by the quantum smooth conditional min-entropy $H_{\min,q}^{\epsilon_s}(\mathbf{O}_n|\mathbf{I}_n, \text{Eve})$[21]. Here, the smoothness error $\epsilon_s$ measures the purified distance between the actual state and an ideal state of $\mathbf{I}_n$, $\mathbf{O}_n$ and $\mathsf{E}$ (see Sect. IV of ref. [28]). Suppose that each trial of an experiment is characterized by the quantum model $\mathcal{Q}$. Denote the QEF with power $\beta_q$ at the $k$'th trial by $F_{q,k}$, which is a function of $I_k$ and $O_k$, and let the variable $T_{q,n}$ be the product of QEFs up to the $n$'th trial, that is, $T_{q,n} = \prod_{k=1}^n F_{q,k}$. In practice, the input at a trial is independent of the outputs of the previous trials given the quantum side information in $\mathsf{E}$ and the inputs of the previous trials. Under this conditional-independence condition,

quantum probability estimation can certify randomness with respect to quantum side information according to the following theorem:

*Theorem 2* (Theorem 3 of ref. [28]): Let $1 \geq \kappa$, $\epsilon_s, p > 0$. Define $\Phi$ to be the event that $T_{q,n} \geq 1/(p^{\beta_q}(\epsilon_s^2/2))$. For each classical-quantum state $\rho_{\mathbf{I}_n\mathbf{O}_n\mathsf{E}}$, either the probability of the event $\Phi$ is less than $\kappa$ or the quantum smooth conditional min-entropy, when the event $\Phi$ happens, satisfies

$$H_{\min,q}^{\epsilon_s}(\mathbf{O}_n|\mathbf{I}_n, \text{Eve}, \Phi) \geq -\log_2(p) + \frac{1+\beta_q}{\beta_q}\log_2(\kappa). \tag{6}$$

The event $\Phi$ can be interpreted as the event that the experiment succeeds. When the experiment succeeds, we compose the quantum smooth conditional min-entropy bound in Eq. (6) with a quantum-proof strong extractor of error $\epsilon_x$ (in trace distance), in order to obtain random bits which are within soundness error (in trace distance) $\epsilon = \epsilon_s + \epsilon_x$ from uniform in the presence of quantum side information. See Sect. V of ref. [28] for the details of the end-to-end randomness generation. Note that an extractor is quantum-proof if it works in the presence of quantum side information. As the TMPS extractor[29,36] is a quantum-proof strong extractor[37], we use this extractor for randomness extraction. The way of running the TMPS extractor for our case is the same as for the case of device-independent randomness generation with respect to quantum side information studied in refs. [15,27,28].

### Constructions of PEFs and QEFs with adversarial imperfections.

Both probability estimation and quantum probability estimation are general frameworks for certifying randomness; however, their implementations are case-dependent as both the classical and quantum models for a trial depend on the case of interest. For the case of device-independent randomness generation, both frameworks have been implemented, see refs. [15,25–28]. In this work we would like to apply probability estimation and quantum probability estimation for randomness generation with partially characterized quantum devices. For this, we need to first construct the classical model $\mathcal{C}$ and the quantum model $\mathcal{Q}$ for an experimental trial in the scenario of our interest, and then construct the corresponding PEFs and QEFs. Below we provide an overview of our constructions. Details are presented in Supplementary Notes 1 and 2.

To construct the models $\mathcal{C}$ and $\mathcal{Q}$ for the scenario of our interest, we observe that although the measurements along the $X$-basis and $Z$-basis are difficult to be precisely characterized, both of them are block-diagonal with respect to various photon-number subspaces. Therefore, the model $\mathcal{C}$ (or $\mathcal{Q}$) can be expressed as a convex combination (or a direct sum) of sub-models $\mathcal{C}_j$ (or $\mathcal{Q}_j$), where the sub-models $\mathcal{C}_j$ and $\mathcal{Q}_j$ are the classical and quantum models conditional on the number of photons $j$ emitted from the source. So, we need only to construct the sub-models $\mathcal{C}_j$ and $\mathcal{Q}_j$ individually, which is discussed in the next two paragraphs.

To construct the sub-models $\mathcal{C}_1$ and $\mathcal{Q}_1$ when a single photon is emitted (i.e., $j = 1$), we take into account of the bounds on the adversarial misalignment and on the adversarial imbalance between the $X$-basis and $Z$-basis, and consider all the possible single-photon states which may be correlated with the side information of Eve. We assume that the measurements in the single-photon subspace are projective, although these measurements are not precisely characterized. So, the misalignment and imbalance are sufficient for characterizing these imperfect measurements. The above assumption can be relaxed to some degree as explained in Supplementary Notes 1 and 2. When Eve can manipulate the misalignment or imbalance depending on the auxiliary degrees of freedom of the single photon such as spatial mode, frequency or polarization, we need to represent the single-photon state and the associated measurement operators in a Hilbert space describing not only the time-bin degree of freedom for information encoding but also the auxiliary degrees of freedom manipulable by Eve. In this case, we take advantage of the assumption that the coherent superposition of states for an auxiliary degree of freedom manipulable by Eve does not play a role throughout the measurement process. (Such assumption has been exploited for verifying entanglement[38] and further for proving the security of quantum key distribution[39] in the presence of side information that can induce detection-efficiency mismatch.) This assumption can be justified if in the setup for time-bin measurements there is no quantum interference between any pair of states for the auxiliary degree of freedom manipulable by Eve (which is true in practice as we think). In addition, the above assumption is consistent with the assumption specified in the Results section that by manipulations Eve can access classical side information but not quantum side information about the measurement performed. Therefore, each measurement operator on a single photon is block-diagonal with respect to various states for the auxiliary degrees of freedom, where each block is described by a qubit measurement. As a consequence, for constructing the sub-models $\mathcal{C}_1$ and $\mathcal{Q}_1$ the single-photon state and the associated measurement operators can be treated without loss of generality as living in a two-dimensional Hilbert space, even in the general case where Eve's manipulations can depend on the auxiliary degrees of freedom of the single photon. We note that for security analysis in the above general case, the bounds on the misalignment and on the imbalance between the $X$-basis and $Z$-basis should be satisfied by the measurement operators in each two-dimensional Hilbert space obtained by projecting onto each particular state for the auxiliary degrees of freedom manipulable by Eve.

On the other hand, when multiple photons are emitted (i.e., $j > 1$) we construct the sub-models $\mathcal{C}_j$ and $\mathcal{Q}_j$ in a device-independent way (i.e., without using any information about the multiphoton state prepared or measurements performed). By the device-independent constructions of sub-models $\mathcal{C}_j$ and $\mathcal{Q}_j$ with $j > 1$, we pessimistically allow Eve's classical or quantum side information to be perfectly correlated with the trial output $O$ given the trial input $I$ and $j > 1$. Consequently, we choose to not certify the randomness contributed by the multiphoton events, and so our security analysis is robust against photon-number splitting attacks. We emphasize that even with the device-independent constructions of sub-models $\mathcal{C}_j$ and $\mathcal{Q}_j$ with $j > 1$, the resulting models $\mathcal{C}$ and $\mathcal{Q}$ still behave well for certifying randomness as the probability of emitting a single photon at each trial is assumed to be bounded from below no matter how Eve manipulates the photon-number distribution.

Once the classical model $\mathcal{C}$ and the quantum model $\mathcal{Q}$ are constructed, we can construct the corresponding PEFs and QEFs. Since the classical model (or the quantum model) for each trial is the identical $\mathcal{C}$ (or $\mathcal{Q}$), we can use the same PEF $F_c(I, O)$ (or the same QEF $F_q(I, O)$) for each trial. According to Theorem 1 (or Theorem 2), the amount of classical (or quantum) $\epsilon_s$-smooth min-entropy in the outputs $\mathbf{O}_n$ certifiable conditionally on the inputs $\mathbf{I}_n$ and on the side information $E$ (or E) is determined by the product $\prod_{k=1}^{n} F_c(I_k, O_k)$ (or $\prod_{k=1}^{n} F_q(I_k, O_k)$). Before the experiment we need to choose a PEF (or a QEF) such that the expected amount of certifiable classical (or quantum) $\epsilon_s$-smooth min-entropy is as large as possible. At the same time, a PEF (or a QEF) satisfies a set of linear constraints imposed by each member of the model $\mathcal{C}$ (or $\mathcal{Q}$). Therefore, we can formulate the constructions of PEFs and QEFs as constrained optimization problems. To solve these optimization problems, we provide effective outer-approximations of the models $\mathcal{C}$ and $\mathcal{Q}$. We note that the outer-approximations of $\mathcal{C}$ and $\mathcal{Q}$ provided by us include the convex closures of $\mathcal{C}$ and $\mathcal{Q}$, respectively. Therefore, in view of the remarks below Eqs. (1) and (5), the constructed PEFs and QEFs can certify randomness even when Eve's side information about the state is classically correlated with Eve's partial knowledge of the input and measurement at a trial.

**Reporting summary**. Further information on research design is available in the Nature Research Reporting Summary linked to this article.

## Data availability
The data that support the findings of this study are available from the corresponding authors upon reasonable request.

## Code availability
The code that produces the results presented in this work is available from the corresponding authors upon reasonable request.

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

## Acknowledgements
We thank Emanuel Knill for stimulating discussions and Ivan Iakoupov for help with running the extractor. This work includes contributions of the National Institute of Standards and Technology, which are not subject to U.S. copyright.

## Author contributions
Y.Z. and H.P.L. contributed equally to this work. Y.Z., H.T., H.P.L., and W.J.M. conceived the original concept and proposed the experiment, which was carried out by H.P.L. together with T.I. and T.H. Y.Z. developed the security-analysis method and conducted the data analysis. The randomness extraction was preformed by Y.Z. and A.M. All authors discussed the results and contributed to the writing of the paper.

## Competing interests
The authors declare no competing interests.

## Additional information

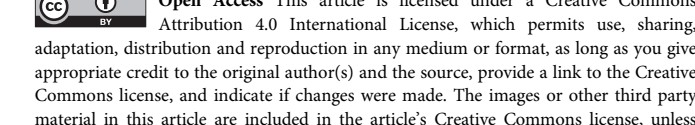

