## [Peer Review File · Nature Communications]

Reviewers' Comments:

Reviewer #2:

Remarks to the Author:

Reading through the revised manuscript and the authors' responses, I found that the authors have made significant revisions. They have further elaborated on the advancements and advantages of the experimental set-up compared to previous similar experiments. The work brings new results to the field of quantum randomness generation, and it is publishable in *Nature Communications*.

However, I find the authors' responses to the previous comments on the physical assumptions and device characterisations in the protocol not satisfactory. To be specific, I am concerned with the issues on the measurement device:

1. In the revised manuscript and the responses, the authors claim that the side channels from the source are considered. To my understanding, the authors now characterise the quantum state emitted from the source as multiple optical modes in a large Fock space. In this case, each photon-number subspace is also defined over multiple modes. However, the descriptions of the measurement process appear to be for the single mode, e.g., Eqs. (S37) and (S38) in the Supplementary Information. The authors should clarify clearly whether side channels are indeed considered for both the source and the measurement device.

2. In the constructions for the (quantum) probability estimation models, the authors have taken the measurements to be projective right from the beginning. It is reasonable to split the measurements for the single photon component from the ones for the multi-photons/zero-photon components, and the measurements for the latter can be treated as projective along the same argument of system dimension dilation in the device-independent cases. However, it is unclear to me why the measurements on the single photon component can be taken as projective. In general, a measurement with binary outputs on a qubit space can still be a POVM. If it is an assumption that the measurements are projective, then it needs to be stated clearly.

3. If the issues above can be clarified, how does an experimentalist effectively verify the misalignments of the measurement angles?

4. The authors have implicitly assumed that the source and the measurement devices are independent. The presented results can be invalid if correlations exist between the source and the measurement device. To my understanding, the classical/quantum adversary in the proposed protocol refers to that the potential eavesdropper is classically correlated with the source, or that she holds the purification. The correlation does not involve the measurement device. The authors should clarify this point.

If the authors can well address these concerns, I would like to recommend the publication of the work in *Nature Communications*.

Reviewer #3:

Remarks to the Author:

The authors have addressed my concerns and I recommend the reviewed version to be accepted in *Nature Communications*.

Summary of the changes made:

- 1) In the Methods section of the main text as well as in the supplementary information, we have added more details on how to handle the side channels in both the state source and the measurement apparatus. This is to address Point 1) raised by Reviewer #2.
- 2) Further in order to address Point 2 and Point 4) raised by Reviewer #2, we clarify the assumptions required by our security analysis in both the main text and the supplementary information.
- 3) In the supplementary information, we provide more details on how to verify or calibrate the imperfections in practice to address Reviewer #2's Point 3.
- 4) Finally we have made a number of additional clarifications and minor corrections in both the main text and the supplementary information to improve the readability of our work.

We note that all changes made are highlighted in blue in the attached PDF copy of our revised manuscript.

REVIEWER COMMENTS:

First of all, we thank all the reviewers for the positive evaluations of our work and for the valuable comments that have helped to improve our manuscript. To express our gratitude to the reviewers for their valuable comments, we now add an acknowledgment in our manuscript. We address all the points raised by Reviewer #2 as follows. We believe that by addressing these points, our manuscript has improved significantly.

Reviewer #2 (Remarks to the Author):

Reading through the revised manuscript and the authors' responses, I found that the authors have made significant revisions. They have further elaborated on the advancements and advantages of the experimental set-up compared to previous similar experiments. The work brings new results to the field of quantum randomness generation, and it is publishable in Nature Communications.

Reply: We thank the reviewer for this positive recommendation.

However, I find the authors' responses to the previous comments on the physical assumptions and device characterisations in the protocol not satisfactory. To be specific, I am concerned with the issues on the measurement device:

1. In the revised manuscript and the responses, the authors claim that the side channels from the source are considered. To my understanding, the authors now characterise the quantum state emitted from the source as multiple optical modes in a large Fock space. In

this case, each photon-number subspace is also defined over multiple modes. However, the descriptions of the measurement process appear to be for the single mode, e.g., Eqs. (S37) and (S38) in the Supplementary Information. The authors should clarify clearly whether side channels are indeed considered for both the source and the measurement device.

Reply: We thank the reviewer for this important comment. In order to avoid confusion with the time-bin mode used for encoding information, we would like to refer to the modes manipulable by Eve as the auxiliary degrees of freedom. By manipulating these auxiliary degrees of freedom, Eve can perform the corresponding side-channel attacks. We agree with the reviewer that in the presence of side channels, the single-photon state and the associated measurement operators need to be represented in a Hilbert space describing not only the time-bin degree of freedom for information encoding but also the auxiliary degrees of freedom manipulable by Eve. In this case, we take advantage of the observation that the coherent superposition of states for an auxiliary degree of freedom manipulable by Eve does not play a role throughout the measurement process because of a lack of an associated, controlled interference effect in the practical setup for time-bin measurements. (Such observation has been exploited for verifying entanglement [Phys. Rev. A 95, 042319 (2017)] and further for proving the security of quantum key distribution [arXiv:2004.04383] in the presence of side channels that can induce detection-efficiency mismatch.) Therefore, each measurement operator on a single photon is block-diagonal with respect to various states for the auxiliary degrees of freedom, where each block is described by a qubit measurement. Consequently, for our security analysis the single-photon state and the associated measurement operators can be treated without loss of generality as living in a two-dimensional Hilbert space. We now clarify this point in the Methods section of the main text as well as in the supplementary information. Accordingly, our method can handle side channels that manipulate both the state and the measurement operators. In addition, we would like to clarify the assumption that by manipulations Eve can access classical side information but *not* quantum side information about the measurement performed at a trial, which is now stated out explicitly in the main text.

2. In the constructions for the (quantum) probability estimation models, the authors have taken the measurements to be projective right from the beginning. It is reasonable to split the measurements for the single photon component from the ones for the multi-photons/zero-photon components, and the measurements for the latter can be treated as projective along the same argument of system dimension dilation in the device-independent cases. However, it is unclear to me why the measurements on the single photon component can be taken as projective. In general, a measurement with binary outputs on a qubit space can still be a POVM. If it is an assumption that the measurements are projective, then it needs to be stated clearly.

Reply: The reviewer has provided an excellent suggestion here. Accordingly, in the Methods section of the main text as well as in the supplementary information we state explicitly the assumption that the measurements on a single photon are projective.

Moreover, we would like to point out that when constructing the (quantum) probability estimation models, our method can deal with general measurements that can be expressed as convex combinations of the assumed projective measurements. We now remark and explain this point in Section I A and Section II A of the supplementary information. The basic reason behind is that our construction of probability or quantum estimation factors works well for both a model and its convex closure, see the current remarks behind Eq. (1) and Eq. (5) in the main text.

At the same locations in the supplementary information, we also note that not all the possible general measurements can be obtained by convex combinations of the assumed projective measurements. How to handle all the possible general measurements with our method deserves further investigation in future work. In addition, we would like to mention that our current method does not allow Eve hold purifications of general measurements, as it assumes that Eve can access classical side information but *not* quantum side information about the measurement performed at a trial.

3. If the issues above can be clarified, how does an experimentalist effectively verify the misalignments of the measurement angles?

Reply: The reviewer has raised an important question here. To address this, we now add general comments in Section III of the supplementary information on how to verify or calibrate the imperfections. In particular, the calibration of the misalignment angle depends on the setup. For the active-selection scheme with polarization encoding, the measurement directions are set by a polarization rotator driven by some electric or magnetic field. Because of the fluctuation of the electric or magnetic field in practice, the measurement directions set by the rotator can vary from trial to trial. Therefore, we need to calibrate the electric or magnetic field applied. For the passive-selection scheme with time-bin encoding as in our experiment, the measurement directions are determined by the splitting ratios of the two beam splitters in the interferometer used as well as by the efficiency mismatch of the two detectors employed. Therefore, we need to calibrate the splitting ratios and the efficiency mismatch. We also clarify the assumption used in our calibration process. Specifically, we assume that the splitting ratios and the efficiency mismatch are stable over time and independent of the auxiliary degrees of freedom manipulable by Eve. In order to handle the cases where the parameters to be calibrated fluctuate over time or can be manipulated by Eve via side-channel attacks, we set conservative bounds on the calibrated imperfections for performing security analysis.

4. The authors have implicitly assumed that the source and the measurement devices are independent. The presented results can be invalid if correlations exist between the source and the measurement device. To my understanding, the classical/quantum adversary in the proposed protocol refers to that the potential eavesdropper is classically correlated with the source, or that she holds the purification. The correlation does not involve the measurement device. The authors should clarify this point.

Reply: We thank the reviewer for this valuable comment. The constructions of the (quantum) probability estimation models presented in our previous manuscript indeed implicitly assume that the state source and the measurement device are independent.

However, the constructions can be easily adapted to the case where the state prepared is classically correlated with the measurement performed at a trial. The basic reason behind is that our construction of probability or quantum estimation factors works well for both a model and its convex closure, see the current remarks behind Eq. (1) and Eq. (5) in the main text. We now clarify this point when we present the explicit constructions of the (quantum) probability estimation models in the supplementary information. At the same time, in the main text we clarify that our method allows classical correlations between the state prepared and the input selected or the measurement performed.

In addition, in the main text we emphasize that our method cannot be applied in the case where at each trial the state prepared is correlated in a quantum manner with the input selected or the measurement performed.

If the authors can well address these concerns, I would like to recommend the publication of the work in Nature Communications.

Reply: We believe that we have been able to address the reviewer's concerns as detailed above.

Reviewers' Comments:

Reviewer #2:

Remarks to the Author:

In the revised manuscript, the authors have addressed my previous concerns. I recommend the reviewed version to be accepted in Nature Communications.

Here, I want to note that the claim in the reply "... the coherent superposition of states for an auxiliary degree of freedom manipulable by Eve does not play a role throughout the measurement process because of a lack of an associated, controlled interference effect in the practical setup for time-bin measurements" is an assumption. In general, Eve's effects could be coherent. In the cases with a detection-efficiency mismatch, such an assumption does not always hold. For example, a detailed discussion can be found in [Quant. Inf. Comput., vol. 9, p. 0131 (2009)]. As an optional comment, I suggest the authors further clarify this point.

Summary of the changes made:

We make necessary changes in both the main text and the supplementary information to address the comment of Reviewer #2.

For convenience, below we include the text of the referee report in black and highlight our reply in blue.

REVIEWER COMMENTS:

Reviewer #2 (Remarks to the Author):

In the revised manuscript, the authors have addressed my previous concerns. I recommend the reviewed version to be accepted in Nature Communications.

Reply: We thank the reviewer for this positive recommendation.

Here, I want to note that the claim in the reply "... the coherent superposition of states for an auxiliary degree of freedom manipulable by Eve does not play a role throughout the measurement process because of a lack of an associated, controlled interference effect in the practical setup for time-bin measurements" is an assumption. In general, Eve's effects could be coherent. In the cases with a detection-efficiency mismatch, such an assumption does not always hold. For example, a detailed discussion can be found in [Quant. Inf. Comput., vol. 9, p. 0131 (2009)]. As an optional comment, I suggest the authors further clarify this point.

Reply: We thank the reviewer for pointing out this. We agree with the reviewer that the statement "the coherent superposition of states for an auxiliary degree of freedom manipulable by Eve does not play a role throughout the measurement process" may be not true in practice. In view of this, we now clarify in both the Methods section and Supplementary Note 3 that the above statement is actually an assumption. Further, we clarify that this assumption can be justified if in the measurement setup there is no quantum interference between any pair of states for the auxiliary degree of freedom manipulable by Eve. In our opinion, the above condition is satisfied in practice. In addition, we clarify in the Methods section that the above assumption is consistent with the assumption stated out in the Results section for our security analysis.

We believe that we have been able to address the reviewer's comment as detailed above.